# Targeting DNA Damage Repair and Immune Checkpoint Proteins for Optimizing the Treatment of Endometrial Cancer

**DOI:** 10.3390/pharmaceutics15092241

**Published:** 2023-08-30

**Authors:** Xing Bian, Chuanbo Sun, Jin Cheng, Bo Hong

**Affiliations:** 1College of Biological and Pharmaceutical Engineering, West Anhui University, Lu’an 237012, China; bian2345@126.com (X.B.); scb19781979@126.com (C.S.); chengjin2005@163.com (J.C.); 2Anhui Province Key Laboratory of Medical Physics and Technology, Institute of Health and Medical Technology, Hefei Institutes of Physical Science, Chinese Academy of Sciences, Hefei 230031, China

**Keywords:** endometrial cancer, DNA damage repair, PARP, immunotherapy

## Abstract

The dependence of cancer cells on the DNA damage response (DDR) pathway for the repair of endogenous- or exogenous-factor-induced DNA damage has been extensively studied in various cancer types, including endometrial cancer (EC). Targeting one or more DNA damage repair protein with small molecules has shown encouraging treatment efficacy in preclinical and clinical models. However, the genes coding for DDR factors are rarely mutated in EC, limiting the utility of DDR inhibitors in this disease. In the current review, we recapitulate the functional role of the DNA repair system in the development and progression of cancer. Importantly, we discuss strategies that target DDR proteins, including PARP, CHK1 and WEE1, as monotherapies or in combination with cytotoxic agents in the treatment of EC and highlight the compounds currently being evaluated for their efficacy in EC in clinic. Recent studies indicate that the application of DNA damage agents in cancer cells leads to the activation of innate and adaptive immune responses; targeting immune checkpoint proteins could overcome the immune suppressive environment in tumors. We further summarize recently revolutionized immunotherapies that have been completed or are now being evaluated for their efficacy in advanced EC and propose future directions for the development of DDR-based cancer therapeutics in the treatment of EC.

## 1. Introduction

Endometrial cancer (EC) is one of the most aggressive malignancies among gynecological cancers in women, with an estimated 417,000 new diagnoses in 2020 globally, and the incidence of EC is rising in over 20 countries, especially for high-grade EC [1,2,3]. The prevalence of established risk factors for EC varies and has been increasing in past decades, including factors such as early menarche, late menopause, nulliparity, menopausal hormone use, obesity and diabetes [4,5].

Endometrial cancers are broadly classified into two subtypes, with the majority of patients (80%) being diagnosed with type I ECs which show a low-grade, endometrioid histology; they are driven by excessive estrogen exposure and associated with a favorable prognosis. In contrast, type II ECs are high-grade cancers, commonly serous and of a clear-cell morphology; they are estrogen-independent and are associated with a poor prognosis, accounting for 40% of EC-related deaths [6,7]. Clinically, type I ECs are often diagnosed early and successfully treated with surgery and/or radiotherapy, whereas serous type II ECs are often diagnosed late and treated with whole-body chemotherapy [2,8]. Unfortunately, tumor recurrence easily occurs and ultimately leads to a five-year survival rate of under 30% in high-grade or serous ECs, and the treatment options for these types of disease are limited.

Previous molecular profiling has shown that the loss of *PTEN* and *PIK3CA* and *PIK3R1* mutations coexist in type I EC, leading to the continuous activation of the oncogenic PI3K-AKT pathway [9,10,11,12]. Other commonly mutated genes in type I ECs include *KRAS*, *FGFR2*, *ARID1A* and *CTNNB1*, along with microsatellite instability, which is found in about one-third of type I ECs [13,14,15]. In contrast, type II ECs are often characterized by alterations in *TP53*, *PIK3CA*, *PPP2R1A* and *FBXW7* [16,17,18]. Currently, molecular characteristic studies including whole-exome sequencing and TCGA’s integrated genome sequencing have classified the endometrioid and serous histology of EC into four distinct subtypes. The first is the *POLE*-mutated (ultramutated) subgroup, and patients of this group exhibit a favorable prognosis. The second is the hypermutated subgroup, which includes microsatellite instability and *MLH1*-promoter hypermethylation. The third and fourth subgroups are characterized by low copy number alterations and high copy number alterations, respectively [9,19,20]. High-copy-number (serous-like) tumors always exhibit a *TP53* gene mutation, and the patients in this group have the worst prognosis. While most low-grade or intermediate-grade endometrioid ECs have favorable prognoses, the majority of high-grade or serous-like ECs always experience recurrence or tumor progression, and these patients ultimately succumb to their disease [19].

Targeting the DNA damage repair system has shown extreme therapeutic potential in a variety of cancer therapies, and the efficacy of this treatment has been linked to the DDR pathway’s ability in cancer cells [21]. However, recent studies have demonstrated that HR activity is competent in the majority of type I ECs, limiting the application of a DDR-based inhibitor in this type of cancer [9,22]. Nonetheless, there have been several studies that have tried to target DNA damage repair proteins, including PARP, which plays an important role in mediating single-strand break (SSB) and double-strand break (DSB) DNA repair in the treatment of EC; the results have demonstrated a modest outcome [23,24]. In contrast, sequencing data have shown that the majority of high-grade or serous ECs are HR-deficient, and some oncogenes such as *MYC* and *CCNE1*, which are inducers of genome instability, are amplified in a subset of serous ECs, causing this type of EC to be dependent on DDR for survival [25,26,27]. Therefore, targeting the DNA damage repair system in EC will provide a rational way of treating EC independent of different histology types [28,29]. In this review, we summarize the preclinical and clinical studies that use DNA repair proteins as therapeutic candidates in treating ECs and discuss biomarkers for predicting the response to these treatments. Since immunotherapy has revolutionized cancer treatment in recent years and antitumor immunity plays a key role in cancer therapy, most recent studies have led to the development of immune checkpoint inhibitors (ICIs) in overcoming the immune escape signals in cancers. Moreover, recent clinical trials that used ICIs as monotherapies or in combination with other agents demonstrated remarkable treatment efficacies in EC patients, especially those with a mismatch repair deficiency. Furthermore, we summarize ICIs in advanced EC and delineate the patients with specific genetic backgrounds that will benefit the most from these ICIs.

## 2. Single-Strand Break Repair

### Base Excision Repair (BER)

Approximately 70,000 DNA lesions arise per cell in the human body each day. If left unrepaired or repaired incorrectly, lesions can convert into deleterious double-strand breaks (DSBs), which represent the most dangerous type of lesion that can threaten the integrity of genome and ultimately result in tumor formation [30,31]. To correctly counteract these different types of lesions, cells have evolved a variety of surveillance mechanisms that respond to different types of lesions. At this point, cells initiate two repair pathways, SSBR (single-strand break repair) and DSBR (double-strand break repair), to repair these lesions. Base excision repair (BER) is the most representative type of SSBR. In a short-patch BER, the first step involves removal of damaged bases by DNA glycosylase to generate an apurinic/apyrimidinic site (AP). Then, the apurinic/apyrimidinic endonuclease 1 (APE1) removes the AP site to induce a single-strand break (SSB). Following the exposure of the SSB, PARP1 recognizes and localizes to exposed SSBs to interact with and recruit a variety of proteins, including DNA polymerase β (polyβ), DNA ligase III (LIG3), XRCC1, ALC1, and PNKP, to mediate the repair process (Figure 1 left) [32,33,34].

SSB repair occurs through the canonical short-patch pathway in which a single damaged nucleotide is replaced. If the short-patch repair is dysfunctional, for example, if POLβ cannot remove the abasic sugar from the 5′terminus, SSBs can be repaired via long-patch BER, in which about 2–30 nucleotides are replaced during repair. In long-patch repair, Polyβ (or Polyδ/ε) induces an extended gap-filling to displace the 5′-terminus and create a flap that is excised by FEN1. Then, DNA ligase 1 ligates 5′ and 3′ nicks (Figure 1, right) [35,36]. Notably, both AP site and SSB are deleterious to the genome and, if left unrepaired, can lead to DSBs. Timely repair of lesions not only ensures only normal cell growth, but also inhibits the growth of cancer cells. Consistently, a recent study reported that germline defects in the BER protein MBD4 leads to multitumor predisposition syndrome [37]. Therefore, understanding the detailed mechanism of BER and its role in regulating cancer is important for developing anti-cancer therapeutics. Indeed, various studies have demonstrated the therapeutic efficacy of targeting BER proteins in cancer treatment [38,39]. However, no study has tested the efficacy of targeting BER proteins in EC.

## 3. DSB Repair

### 3.1. Homologous Recombination (HR) Repair

If left unrepaired or repaired incorrectly, SSBs are converted into DSBs, which are the most deleterious forms of DNA damage that lead to the loss of the majority of chromosomal regions [40,41]. Therefore, accurate reorganization and repair of DSBs are essential to maintaining genome integrity and preventing the formation of cancer. DSBs are mainly repaired using two pathways: non-homologous end joining (NHEJ) and homologous recombination (HR) [42,43,44,45]. In the HR pathway, the first step involves the initiation of DNA end resection at the break site to expose the long stretches of 3′ single-strand DNA (ssDNA) (Figure 2). Proteins that mediate DNA end resection include the MRE11-RAD50-NBS1/XRS2 (MRN/X) complex, CtIP, and BRCA1. Once the DNA ends are resected, replication protein A (RPA) efficiently binds to ssDNA to protect it from nucleases and further prevent the formation of secondary structures via the self-annealing of the ssDNA. RAD51 is then recruited to ssDNA sites to form RAD51 nucleoprotein filaments and ensure invasion of the long stretches of the 3′ ssDNA into homologous duplex DNA (Figure 2) [46,47]. Through this process, genetic information is accurately transferred to daughter cells. A study reported that germline mutations in HR-related genes, such as *BRCA1*, *BRCA2*, *RAD51D,* and *PALB2*, led to the development of EC in a subset of patients [48].

### 3.2. Classical Non-Homologous End Joining Repair (c-NHEJ)

In mammalian cells, DSBs that occur during the cell cycle are repaired predominantly via the classical non-homologous end joining (c-NHEJ) pathway. c-NHEJ-mediated repair involves the direct end-to-end ligation of DSB ends [49]. A biochemical analysis has demonstrated that blunt-end ligation without resection is mediated by the Ku-XRCC4-DNA ligase IV complex (Figure 3). In contrast, ligation of incompatible DNA ends is strongly stimulated by the DNA-PKcs-Artemis complex, which removes 5′ and 3′ DNA overhangs through its endonuclease activity to create DNA ends that can be ligated by the XRCC4-ligase IV complex [50,51]. Notably, XLF and PAXX were recognized as new NHEJ factors involved in ligase-complex-mediated DNA end ligation (Figure 3).

### 3.3. Alternative Non-Homologous End Joining (A-NHEJ)

When the c-NHEJ pathway is compromised, A-NHEJ is activated, which involves much more extensive resection of the DNA ends. A-NHEJ is also known as microhomology-mediated end joining (MMEJ) and is mostly active in the S and G2 phases of the cell cycle. A-NHEJ requires 2–20 bp microhomology [52]. The proteins involved in A-NHEJ include Pol θ, PARP1, CtIP and the MRN complex. The first step of A-NHEJ is the PARP1-mediated localization of the CtIP–MRN complex to the DNA ends [53]. Phosphorylated CtIP promotes activation of the endonuclease function of the MRN complex, which initiates resection by generating 15–100 nucleotide 3′overhangs that are recognized as microhomology sequences of the ssDNA [54]. Then PARP1, MRN complex, and Pol θ promote the alignment of ssDNA via microhomology sequences. Notably, when the annealed microhomologies are embedded within the long 3′ ssDNA, Pol θ removes non-homologous 3′ ssDNA, and the ERCC1 and XPF nucleases digest non-homologous 3′ ssDNA sequences. Finally, Pol θ mediates DNA synthesis to fill the gaps and the LIG3-XRCC1 complex is utilized to repair the remaining nicks [55,56].

## 4. Single-Strand Annealing (SSA)

Single-strand annealing (SSA) has more in common with A-NHEJ than c-NHEJ, because both SSA and A-NHEJ require extensive resection of DNA ends to realize microhomology [57,58,59]. Compared with A-NHEJ, SSA needs more homologous sequence exposure, and the 3′ ssDNA stretches created by the MRN complex and CtIP are further resected by nuclease EXO1, BLM, or DNA replication helicase/nuclease 2 (DNA2) to generate long stretches of 3′ ssDNA (Figure 3) [60,61,62]. The long stretches of 3′ ssDNA are stabilized via RPA binding. In contrast to HR, SSA is RAD51-independent and easily generates deletions and translocations (Figure 3). Notably, before ligation, the non-homologous 3′ ssDNA stretches must be resected and removed via nucleotide the excision repair complex XPF-XRCC1 and mismatch repair (MMR) complex MSH2-MSH3 [49,57,63].

## 5. Targeting DNA Damage Repair Proteins in EC

Most low-grade and early-stage endometrioid tumors have excellent prognoses and >95% five-year survival rate; however, high-grade or serous tumors have extremely poor outcomes as a result of chemoresistance. According to TCGA datasets, 25% of high grade endometrioid tumors and serous tumors exhibit widespread copy number alterations [9,17,26]. The genomic characteristics of serous tumors are common with those of high-grade serous ovarian cancers and triple-negative breast cancers. In high-grade and serous EC, mutations in DNA damage repair genes and high levels of genome instability are frequent; therefore, targeting one or more DNA damage repair proteins is a rational strategy for treating advanced EC [64,65]. The following sections summarize preclinical studies that have targeted DNA damage proteins for treating EC. Moreover, therapeutic targets are suggested to optimize the treatment efficacy of targeted therapy and immunotherapy in EC.

### 5.1. Targeting PARP in EC

According to one study, PARP1 protein is overexpressed in the majority of cases of EC [66]. Targeting PARP with PARP inhibitors represents a promising therapeutic strategy for treating EC. Indeed, many studies have tested the efficacy of PARP inhibitors in EC with different genetic backgrounds, including PTEN-deficient and PTEN-WT. Dedes et al. reported that the PARP inhibitor KU0058948 effectively inhibited PTEN-deficient endometrioid EC [67]. Consistently, Philip et al. reported that two PTEN-deficient EC cell lines were more sensitive to the PARP inhibitors olaparib and talazoparib than PTEN-WT cells [68]. By contrast, according to Miyasaka et al., the efficacy of the PARP inhibitor olaparib was not associated with PTEN status, because only 3/12 PTEN-deficient EC cell lines were sensitive to olaparib [23]. In line with this study, our previous study also demonstrated that PTEN-deficient endometrioid EC cells were not responsive to the PARP inhibitor olaparib in vitro and in vivo [69]. Janzen et al. demonstrated that the anti-cancer activity of olaparib in EC was negatively associated with a low estrogen microenvironment in vivo [24]. A 42-year-old woman with recurrent low-grade endometrioid EC and germline *BRCA2* mutation was treated with olaparib. Because the patient responded robustly to the olaparib, PARP inhibitors may be effective in EC with deleterious *BRCA1/2* mutations, especially in patients with a biallelic inactivation of *BRCA1/2* [70]. Consistently, in ovarian cancer, patients with biallelic *BRCA1/2* mutations showed improved outcomes than those with monoallelic *BRCA1/2* mutations [71]. Two studies reported that ~20% of serous ECs harbor HR defects and may be sensitive to PARP inhibitors [25,27]. Consistently, a recent study used a patient-derived xenograft (PDX) model to examine genomic heterogeneity in EC. Given that the established PDX models include all EC molecular subtypes, the study investigated the efficacy of PARP inhibitor in these models and found that the high-copy-number (serous-like) molecular subtype was sensitive to the PARP inhibitor, which was independent of PTEN status, because patient-derived xenograft modes with PTEN mutations were unresponsive to talazoparib [22]. Thus, studies must be performed to evaluate PTEN status in determining response of PARP inhibitors in EC, and PARP inhibitors may not be sufficient as monotherapy in treating EC.

Several preclinical studies have tested PARP inhibitors in combination with other drugs to optimize treatment efficiency [72]. We previously reported that endometrioid EC models are not responsive to the use of PARP inhibitor as monotherapy, but show superior sensitivity to compound PARP-PI3K inhibition, due to the reduction in HR activity following PI3K inhibition. Our study thus provides a rational combination strategy to effectively treat endometrioid EC [69]. Consistently, Philip et al. reported that an additional PI3K inhibitor, BKM120, enhanced the efficacy of the PARP inhibitors olaparib or talazoparib in EC cells with PTEN deficiencies by inhibiting the formation of RAD51 foci [68]. A phase Ib study used AKT and PARP inhibitors to treat ovarian, endometrial, and breast cancers. The combination of AKT and PARP inhibitors had no serious adverse effects and elicited a prolonged response in ovarian, endometrial, and breast cancers, with especially superior activity in EC. This phase Ib study further recommended a phase II trial of capivasertib and olaparib in patients with advanced EC [73]. *HER2* is overexpressed in ~30% of serous ECs. A recent study used uterine serous carcinoma cell lines to test the efficacy of a PARP inhibitor as monotherapy and in combination with the HER2 inhibitor neratinib. The combination of olaparib and neratinib acted synergistically against *HER2*-overexpressing and HR-proficient serous endometrial carcinoma [74]. Moreover, whether PTEN deficiency predicts the sensitivity of EC cells to PARP inhibitors is controversial. A study used a panel of PTEN-deficient EC cell lines to test the PARP inhibitor olaparib as monotherapy or in combination with an inhibitor of pBADS99 phosphorylation. The data in this study demonstrated that the PTEN-deficient EC cells were responsive to olaparib, and combined treatment with olaparib and pBADS99 phosphorylation inhibitor NPB resulted in synergistic antitumor effects in the PTEN-deficient ECs in vitro and in vivo. Mechanistic analysis showed that NPB treatment increased DNA damage and reduced the activity of HR, thereby improving the efficacy of the PARP inhibitor in EC with PTEN deficiency [75]. Interestingly, a recent study showed that ATAD5 deficiency can sensitize cancer cells to PARP inhibition. The study examined the cBioPortal database and found that ~10% of ECs are ATAD5-deficient; thus, ATAD5 mutation status might be a biomarker for predicting the sensitivity of PARP inhibitor in treating EC [76]. Taken together, PARP inhibitors may have an indispensable role in the management of EC and should be considered in combination with other treatments.

### 5.2. Targeting ATR-CHK1 Signaling in EC

EC is characterized by genome instability with replication stress (RS) due to PTEN deficiency or P53 mutation. In this genetic background, cancer cells may survive by activating the DNA damage repair and replication stress response (RSR) pathways [77]. ATR-CHK1 signaling is activated in response to RS and DSBs. In response to RS, the ATR-CHK1 axis acts as the main transducer of RSR by inhibiting late origin firing and arresting the cell cycle in the S phase. In addition, ATR-CHK1 signaling can respond to RS by (1) increasing deoxynucleotide synthesis, (2) promoting the dormant origin firing within the stalled replication fork and (3) stabilizing and resolving the stalled replication fork for a timely restart of replication [78,79]. Therefore, targeting ATR-CHK1 signaling is an attractive therapeutic strategy for the treatment of cancer with high genome instability [80]. In terms of EC, a study reported that the use of ATR or CHK1 inhibitor as monotherapy enhanced the sensitivity of EC cells to DNA-damaging agents, including cisplatin and doxorubicin [29]. Moreover, the combined use of ATR and CHK1 inhibitors synergistically induced DNA damage and inhibited cell proliferation in EC cells. Serous EC is associated with chemoresistance and poor survival. DNA sequencing data have revealed that the *CCNE1* oncogene is amplified in ~50% of serous and 8% of endometrioid ECs, respectively [27,81]. A recent study by Xu et al. investigated the efficacy of an ATR inhibitor in treating EC with a *CCNE1* amplification. The data indicated that the induction of *CCNE1* led to activation of ATR signaling, and the use of an ATR inhibitor can reverse the inherent resistance of the WEE1 inhibitor. Moreover, the combination of a low dose of an ATR inhibitor and a WEE1 inhibitor synergistically reduced cell viability and colony formation and increased replication fork collapse in EC cells with *CCNE1* amplifications. A further biomarker analysis indicated that *CCNE1* copy number was a clinically tractable biomarker for predicting the sensitivity of EC to ATR-CHK1 inhibition [82].

### 5.3. Targeting WEE1 in EC

It is well recognized that almost all serous ECs occur with a *TP53* mutation which plays an important role in G1/S and G2/M cell-cycle checkpoint regulation in response to DNA damage, thereby permitting normal and cancer cells to repair damaged-DNA before entering the S and G2 phases. P53 mutations and deficiencies in cancer cells are highly dependent on the WEE1 kinase to prevent aberrant G2/M cell-cycle entry and mitotic catastrophe [83]. WEE1 serves as a key regulator of genome integrity via inhibiting CDK1/2-mediated DNA replication during the G1-S cell-cycle transition [84]. Therefore, targeting the G2/M checkpoint via suppressing WEE1 is an attractive strategy for targeting a P53-mutant or -deficient cancer. Recent studies have shown the therapeutic efficacy of targeting WEE1 in EC. According to Meng et al., monotherapy with the WEE1 inhibitor AZD1775 was effective against P53-mutant EC via the induction of cell apoptosis. AZD1775 treatment enhanced the efficacy of a PARP inhibitor, and the combined use of AZD1775 and olaparib resulted in synergistic antitumor effects in EC cells with a P53 mutation [85]. Consistently, a recent clinical phase I trial of AZD1775 demonstrated its activity in EC, and a biomarker analysis further revealed that the expression of *CCNE1* mRNA predicted the sensitivity of EC to AZD1775 [86]. The molecular characterization of serous EC revealed frequent cell-cycle dysregulation and a high level of oncogene-induced RS. Indeed, a high level of DNA replication stress is vulnerable to the inhibition of WEE1. In a phase II study, AZD1775 exhibited robust activity in recurrent serous EC, with an ORR of 29.4% and a progression-free survival (PFS) rate of 47.1% [87]. However, the evaluation could not identify a biomarker that predicted sensitivity to AZD1775. Nonetheless, co-alterations in multiple pathways and alternative measurements of RS are of interest to identify patients with serous EC who may benefit from WEE1 inhibition. Overall, this clinical study suggests that targeting WEE1 is a promising therapeutic modality for serous EC.

### 5.4. Ongoing Clinical Trials

Based on the efficacy of DDR inhibitors in preclinical studies, various clinical trials are evaluating the efficacy of DDR inhibitors in EC, including the inhibitors of PARP, CHK1, and WEE1 (Table 1). A PARP inhibitor is under evaluation in different stages of clinical trials (Table 1). A study of the CHK1 inhibitor BBI-355 is recruiting EC patients with EC to evaluate its efficiency in the treatment of EC (NCT05827614). Additionally, two clinical trials are recruiting patients to evaluate the efficacy of an ATR inhibitor in combination with chemotherapy or the use of a PARP inhibitor in EC (NCT04491942 and NCT03682289).

## 6. DDR Correlates with Cancer Cell Immunogenicity

The DDR system plays key roles in maintaining the genomic integrity of cancer cells. Dysregulating DNA repair processing or uncontrolled activity of repair factors can result in DNA damage, thereby promoting genome instability in cancer cells. In recent years, targeting DDR has become an attractive therapeutic modality for cancer. However, DDR inhibitors are cytotoxic, and studies have reported induction of the innate and adaptive immune response after radiotherapy or chemotherapy. Studies in the last decades have shown that the DDR impacts several aspects of tumor cell immunogenicity, leading to the immune evasion of cancer cells [88,89]. Tumor immunogenicity is the ability of the immune system to recognize and eliminate cancer cells and mainly includes three parameters: antigenicity, adjuvanticity and reactogenicity. All these three parameters work together and define the ability of the immune system to recognize and eliminate cancer cells. DDR-mediated regulation of tumor immunogenicity is complex and involves the activation of multiple pathways, including the innate cGAS-STING immune pathway, which is activated by cytosolic DNA, generated through DNA damage induced by endogenous or exogenous factors (Figure 4) [90]. Moreover, the exogenous inhibition of DDR or inherent DDR defects have been reported to induce expression of immune checkpoint proteins in cancer cells, for example, increased PD-L1 expression in MMR-deficient (MMR-D) cells compared with MMR-proficient (MMR-P) cells (Figure 4) [88]. Under these conditions, checkpoint proteins, often activated by DDR inhibitors, are repurposed to jeopardize theantitumor immune response by binding with their ligands or receptors, and therefore, create an immune-suppressive tumor microenvironment. Thus, to circumvent these immune-suppressive effects, extensive efforts have focused on identifying potential combination therapies to overcome resistance mechanisms and reactivate the activity of T-cells toward the tumor.

## 7. Exploiting the Interplay between DDR and Immunity for Cancer Therapy

Recent studies suggest that pharmacological manipulation of DDR can enhance the efficacy of immunotherapy in cancer treatment. A large number of studies have shown that the implementation of DDR inhibitors elicits innate and adaptive immune response in cancer cells, leading to the immune evasion of cancer cells. Thus, the combination of DDR inhibitors with ICIs showed synergistic antitumor efficacy in cancer therapy, including ovarian cancer and breast cancers [91,92]. Immunotherapy currently represents an attractive strategy for patients with EC, mainly those with advanced or recurrent disease, with no effective treatment options available after standard chemotherapy [93]. Given the high expression of PD-1/PD-L1 in EC, it is reasonable to test inhibitors targeting the PD-1-PD-L1 axis to treat EC [94]. Indeed, many studies have tested the efficacy of PD-1 or PD-L1 antibodies in different molecular subtypes of ECs. More importantly, a combination of DDR agents with immunotherapy may expand the therapeutic antitumor effects in EC.

### 7.1. Pembrolizumab

The anti-PD-1 antibody has shown prolonged and beneficial responses in various human cancers, including EC. A study reported that cancers that carry mutations in *POLE* might serve as good candidates for ICIs. In a recent clinical trial, the efficacy of the PD-1 antibody pembrolizumab was tested in 75 patients with advanced EC. The results showed that a subset of patients was responsive to pembrolizumab. Of the three patients with complete response to pembrolizumab, one had *POLE* mutation, one exhibited microsatellite stability (MSS), and one had unknown microsatellite instability (MSI) status (Table 2) [95]. The findings thus pave the way to utilize the anti-PD-1 antibody in the treatment of EC. Further, in a clinical trial, the efficacy of pembrolizumab was tested in 244 patients with MSI or MMR-D, including 47 patients with EC. The results of this study showed that the majority of patients had a reduction in tumor size upon pembrolizumab treatment, especially 8 patients had complete response [96]. A longer follow-up study in patients with MMR-D or MSI-high (MSI-H) cancer demonstrated that pembrolizumab treatment had clinically meaningful effects, as well as a manageable safety profile [97]. Therefore, pembrolizumab treatment has specific effects in EC with MSI or MMR-D. Indeed, pembrolizumab was approved by the Food and Drug Administration (FDA) in 2017, for treating MSI or MMR-D solid tumors [98]. Notably, although MSI or MMR-D occur in one-third of the patients with EC, it has different molecular subgroups. Results from a clinical phase II trial revealed that patients with Lynch-like syndrome, who had high tumor mutation burdens, were highly responsive to pembrolizumab, and only 44% of the patients with sporadic MSI-H responded to pembrolizumab [99]. Consistently, a phase II trial showed that patients with EC with mutational MMR-D had higher response rates to PD-1 inhibitor pembrolizumab [100]. Together, these studies support the use of pembrolizumab in treating patients with EC, who have high tumor mutation burdens.

### 7.2. Nivolumab

The initial clinical trial of the human monoclonal antibody nivolumab in EC was implemented in 2016 and included two patients. WES results indicated that both patients had *POLE* and *MSH6* mutations. According to the RECIST criteria, both patients showed significantly reduced tumor burdens, following a dose of 3 mg/kg of nivolumab every 2 weeks (Table 2) [101]. A phase II study tested the efficacy of 240 mg of nivolumab every 2 weeks in patients with solid tumors, including EC. Patients who had complete response had MSI-H, further indicating the importance of MSI status in predicting treatment response [102]. Because nivolumab showed activity in colon cancer with MMR-D, a recent clinical trial tested the efficacy of nivolumab in non-colorectal cancer with MMR-D. The results showed that nivolumab has promising activity in non-colorectal MMR-D cancers, especially in endometrioid EC [103].

### 7.3. Dostarlimab

A phase I nonrandomized trial has completed evaluating the efficacy of dostarlimab in recurrent EC that progressed after platinum-based chemotherapy. Patients with EC who achieved an objective response were deficient in the MMR system (Table 2) [104]. Further, objective response rate (ORR) was equal to 44.7% in the MMR-D EC compared with an ORR =3.7% in MMR-P/MSI EC. Recently, the interim results from GARNET—a phase I, single-arm study—revealed that dostarlimab exhibited robust antitumor activity in both MMR-D/MSI and MMR-P/MSS EC, as well as a manageable safety profile [105]. Based on the results of this study, the FDA approved the use of dostarlimab in advanced EC with MMR-D [106].

### 7.4. Durvalumab

The anti-PD-L1 efficacy of durvalumab was evaluated in the phase II PHAEDRA trial. The results revealed a PFS of 8.3 months in MMR-D versus 1.8 months in MSS, demonstrating that in patients with EC, the antitumor effects of durvalumab were dependent on MMR status [107]. The results of a recent clinical trial of durvalumab combined with olaparib in treating metastatic or recurrent EC showed that the combined use of durvalumab and olaparib was well tolerated, but did not meet 50% 6-month PFS in advanced EC [108].

### 7.5. Other Anti-PD-L1 Antibodies

In a phase I trial, the efficacy of an anti-PD-L1 antibody atezolizumab was evaluated in 15 patients with EC. Patients with a clinical response to atezolizumab were MMR-D (Table 2) [109]. In 2019, the efficacy of human anti-PD-L1 antibody avelumab was evaluated in patients with EC [110]. The results showed an ORR of 40% in the MMR-D group; however, these responsive patients were absent for PD-L1 expression, indicating that further studies are needed to identify biomarkers that predict sensitivity to ICIs.

### 7.6. Immunotherapy in Combination with Chemotherapy

Because DNA damage agents enhance the tumor mutation burden and induce neoantigen exposure, the use of chemotherapeutic agents in combination with ICIs represents a promising strategy to enhance treatment efficiency in patients with cancer. A phase II study of pembrolizumab with chemotherapy in EC demonstrated remarkable efficacy [111]. Consistently, a phase III study showed that a combination of pembrolizumab with carboplatin or paclitaxel resulted in significantly longer PFS than chemotherapy alone [112] (NCT03914612). A phase II trial has completed patient recruitment and is clinically evaluating the efficacy of pembrolizumab combined with doxorubicin in advanced, recurrent, and metastatic EC (NCT03276013). Similarly, a clinical trial is evaluating the efficacy of a combination of atezolizumab with carboplatin–paclitaxel in advanced or recurrent EC (NCT03603184). A phase III study is evaluating the combined effects of dostarlimab and carboplatin or paclitaxel in EC (NCT03981796). If positive, the findings of these studies can improve treatment options for patients with EC, regardless of their tumor stage. Consistently, a recent phase Ib/II biomarker-driven study is evaluating suitable combination agents with atezolizumab based on sequencing data from patients with persistent or recurrent EC (https://clinicaltrials.gov/ct2/show/NCT04486352 (accessed on 24 July 2020)).

**Table 2 pharmaceutics-15-02241-t002:** Clinical studies of the use of ICIs in monotherapy for EC.

Immune Checkpoint Inhibitor	Patient Number	Response Biomarker	Phase	Status	Reference
Pembrolizumab	75	*POLE*-mutation and MSI	Completed	Completed	[98]
Pembrolizumab	47	MSI or MMR-D	Phase I	Recruiting	[99]
Pembrolizumab	25	MSI-high	Phase II	Active, not yet recruiting	[101]
pembrolizumab	24	MMR-D	Phase II	Active, not yet recruiting	[102]
Nivolumab	2	*POLE* and *MSH6* mutation	unknown	Unknown	[103]
Nivolumab	2	MSI-high	Phase II	Unknown	[104]
Nivolumab	13	MMR-D	Phase II	Active, not yet recruiting	[105]
Dostarlimab	104	MMR-D	Phase I	Recruiting	[106]
Dostarlimab	75	MMR-D	Phase I	Recruiting	[107]
Durvalumab	71	MMR-D	Completed	Completed	[109]
Atezolizumab	15	MMR-D	Completed	Completed	[111]
Avelumab	33	MMR-D	Phase II	Active, not yet recruiting	[112]

## 8. Immunotherapy in Combination with PARP Inhibitors

Collectively, the clinical results of the use of PD-1 or PD-L1 antibodies in patients with EC seem to positively correlate with the MMR-D or MSI status, and the efficacy of the PD-1 or PD-L1 antibodies in MMR-P EC remains modest. However, a clinical trial suggested that microsatellite-stable tumors with high tumor mutation burdens may benefit from immunotherapy; thus, high tumor mutation burden, but not MMR-D or MSI status, may determine the therapeutic efficacy of ICIs [113].

Combination therapy with targeted agents might increase the response rate to ICIs. Studies have shown that treatment with PARP inhibitors resulted in impaired DNA damage repair, increased tumor mutation burden, increased neoantigen exposure, and increased immune recognition of cancer cells [114,115]. In this context, a phase II study is investigating the efficacy of atezolizumab, bevacizumab, and rucaparib as monotherapy or in combination, in patients with recurrent EC (NCT03694262). Another trial is clinically investigating the efficacy of durvalumab with or without olaparib as maintenance therapy after first-line treatment in advanced or recurrent EC (NCT04269200). In addition, two combination therapies, including dostarlimab + niraparib and nivolumab + rucaparib, are now being investigated in advanced, recurrent, or metastatic EC (NCT03016338 and NCT03572478, respectively). 

## 9. Conclusions

Although early-stage or low-grade EC have good prognoses, high-grade serous EC is highly aggressive and accounts for 40% of the deaths. Cytotoxic chemotherapy alone or in combination with radiotherapy remains the treatment of choice for high-grade serous EC. Identifying new therapeutic strategies to enhance treatment efficacy in patients with EC is an urgent clinic need. A study has demonstrated that the overexpression of DNA damage repair genes in EC was positively correlated with poor prognosis [66,116]. In terms of high-grade and serous EC, the HR repair system is deficient and tumors possess high levels of genome instability due to the amplification of oncogenes, such as *CCNE1* and *MYC*. Therefore, modulating the repair system, such as by boosting RS or inhibiting DDR, can provide effective treatment options in serous EC. The modest efficacy of PARP inhibitors as monotherapies in EC highlights the need for combination therapy to improve the efficacy of PARP inhibitors. Targeting ATR-CHK1 signaling and WEE1 had antitumor effects in a preclinical model of EC. Strategies that improve the treatment efficacy of ATR or CHK1 inhibitors should be provided more attention to, especially in a serous histology setting. Furthermore, given the molecular heterogeneity of EC and the risk of toxicity based on DDR-targeting therapies, great attention is required to define optimal combination modality, dose concentration, and schedule of these agents. Because immune stimulation has been detected after use of DDR inhibitors in preclinical and clinical studies, understanding the immune microenvironment, in which DNA lesions can activate antitumor immunity, is important. For EC, combined use of DDR inhibitors with ICIs is now being tested in the clinic, and the results are awaited. In conclusion, understanding of the biology of EC will facilitate the development of rational combination modalities to prevent resistance to DDR-based target therapy, ultimately leading to prolonged survival in patients with EC.

## 10. Future Directions and Perspectives

The combination of pembrolizumab and lenvatinib has recently been approved for use in EC, regardless of different genetic background, including MSI and MSS or MMR-P. However, some patients do not respond well to this treatment due to primary or acquired resistance, as well as treatment-associated toxicity [117]. The use of DNA damage agents as monotherapy or in combination with chemotherapy has shown efficacy in small numbers of ECs, highlighting the need to identify effective treatment strategies for ECs. Targeting DNA damage repair systems, such as HR and RSR, or modulating the cell cycle is of particular interest in EC, especially with high-grade or serous EC. To our knowledge, various factors, including epigenetic regulators, such as BET family members, transcription factors, as well as some non-coding RNAs, regulate HR or RSR [118,119]. A better understanding of factors involved in DNA repair and identification of their targets is needed to develop and apply combination therapies in EC.

FDA has recently approved a combination of immunotherapy for treating EC; however, only a subset of patients that will likely benefit from it. Most patients are not responsive to immunotherapy because of the immunosuppressive tumor microenvironment. The focus is shifting toward the use of other strategies, such as DNA damage agents that expose more neoantigens to enhance the efficacy of ICI, to reduce tumor recurrence in cancer treatment. Indeed, the therapeutic efficacy of a combination of PARP inhibitors and ICI is being clinically evaluated in patients with EC, and the results are awaited [120]. Overall, preclinical and clinical studies must focus on improving treatment efficacy to benefit all molecular types of EC regardless of therapy-type-DNA damage-based targeted therapy or a combined immunotherapy.

## Figures and Tables

**Figure 1 pharmaceutics-15-02241-f001:**
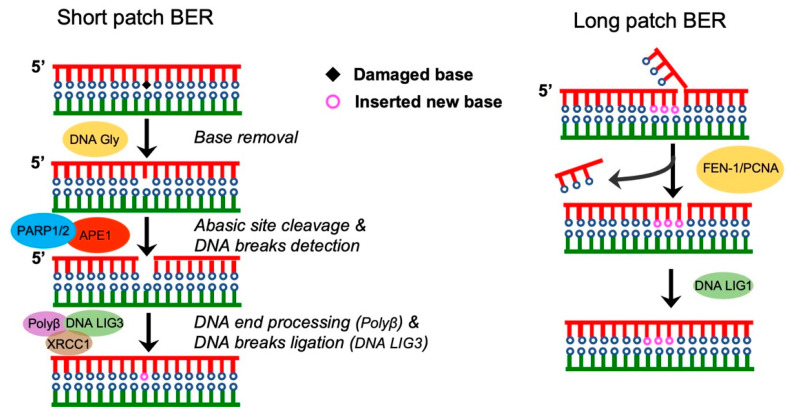
The repair of single-strand DNA breaks via BER. (**Left**: short-patch BER) The damaged bases are removed by DNA glycosylase, then the remaining abasic site (Ab) is cleaved by an apurinic/apyrimidinic endonuclease, leaving a 1-bp DNA gap. The 5′ abasic sugar is removed by DNA polymerase β (POLβ), which inserts a new nucleotide into the DNA gap. Finally, the nick is ligated by DNA ligase 3 (LIG3). (**Right**: long-patch BER) In long-patch repair, about 2–30 nucleotides are replaced. Polyβ induces the formation of an extended gap, then the gap displaces the 5′-terminus to create a flap that is excised by FEN1. Finally, the DNA ligase 1 ligates 5′ and 3′ nicks.

**Figure 2 pharmaceutics-15-02241-f002:**
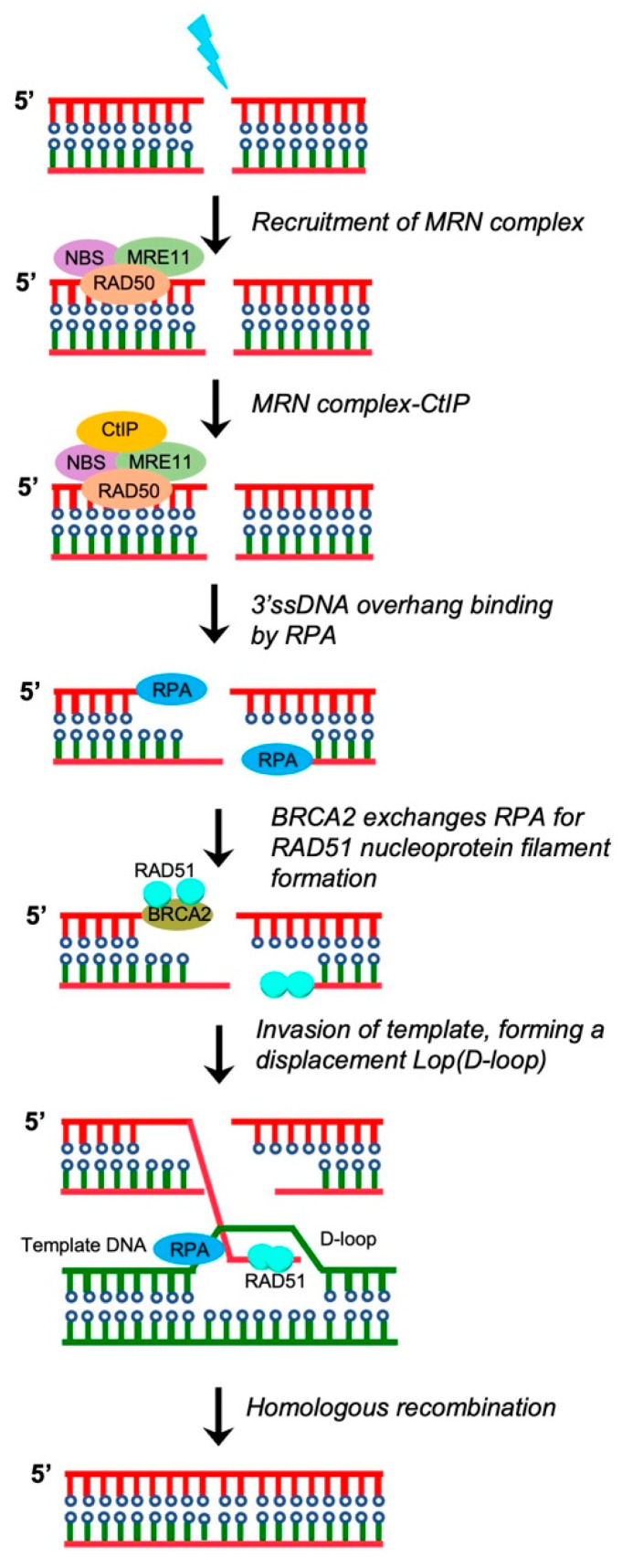
DNA double-strand breaks by homologous recombination. The first step for homologous recombination includes the initiation of end resection, which is mediated by the CtIP—MRN complex (short-patch) or the EXO1/DNA2 (long-patch) nuclease. DNA end resection leads to the generation of 3′ssDNA overhangs, which are bound by replication protein A. Then, BRCA2 exchanges replication protein A on the DNA ends to promote the formation of RAD51 nucleoprotein filaments. The RAD51–ssDNA nucleoprotein filaments mediates homology search by invasion of template dsDNA. Furthermore, the RAD51–ssDNA nucleoprotein filaments form a synaptic complex that contains a three-stranded DNA helix intermediate. In this process, DNA polymerase δ (Pol δ) plays an important role in the synthesis of nascent strands.

**Figure 3 pharmaceutics-15-02241-f003:**
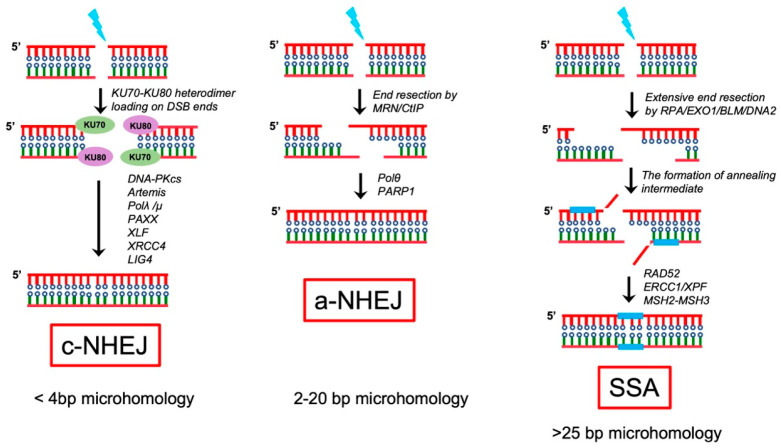
Non-homologous end joining repair mediates the repair of DNA double-strand breaks. DNA double-strand breaks can be repaired via c-NHEJ (**left**), A-NHEJ (**middle**), and single-strand annealing (SSA) (**right**). 53BP1 is a positive regulator of c-NHEJ, and the process usually requires 4-bp microhomology. A-NHEJ, also called microhomology-mediated end joining, requires 2–20 bp microhomology for break repair. PARP1 and Pol θ are important for A-NHEJ. Higher levels of resection can further promote SSA repair pathway, which requires >25 bp microhomology. BLM and EXO1 account for additional resections in SSA. Replication protein A binds and stabilizes ssDNA and promotes SSA. RAD52-mediated annealing of a homologous sequence is important for SSA. ERCC1/XPF cuts the remaining 3′ nonhomologous ssDNA prior to ligation by DNA LIG1.

**Figure 4 pharmaceutics-15-02241-f004:**
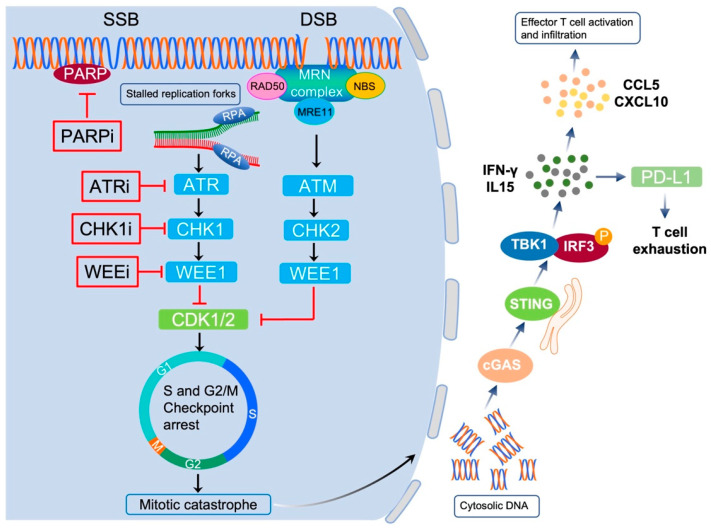
DNA damage repair blockade stimulates innate antitumor immunity via the cGAS-STING pathway. Upon DNA damage, DNA repair proteins are recruited to DNA damage sites for DNA repair. Inhibition of components of the repair pathway leads to cell-cycle checkpoint abrogation and inappropriate mitotic entry, and ultimately, induces mitotic catastrophe. In addition to being cytotoxic, DDR inhibitors exhibit antitumor immunity. PARPi, ATRi, or CHK1i-induced DSBs generate cytosolic dsDNA fragments, which activate the cGAS-STING innate immune pathway to initiate the IFN-γ response. This innate immune response upregulates chemokines, such like CCL5 or CXCL10, to enhance T cell recruitment. Moreover, PD-L1 expression is upregulated by IFN-γ that may lead to T cell exhaustion, an effect that can be abrogated using PD-1 or PD-L1 antibodies.

**Table 1 pharmaceutics-15-02241-t001:** Ongoing clinical trials of the use of DDR inhibitors in EC.

Inhibitor	Combination with	Phase	Status	NCT Number
Olaparib	Monotherapy	Phase I	Not yet recruiting	NCT05320757
Olaparib	CYH33	Phase I	Recruiting	NCT04586335
Niraparib	Monotherapy	Phase II	Recruiting	NCT04716686
Rucaparib	Nivolumab	Phase I and II	Terminated	NCT03572478
Niraparib	TSR-042	Phase II	Active	NCT03016338
Olaparib	Carboplatin	Phase I	Completed	NCT01237067
Olaparib	Selumetinib	Phase II	Recruiting	NCT05554328
AZD5305	Paclitaxel or Carboplatin or T-Dxd or Dato-Dxd or Camizestrant	Phase I and II	Recruiting	NCT04644068
Olaparib	AZD2014 or AZD5363	Phase I and II	Active	NCT02208375
Rucaparib	Bevacizumab	Phase II	Active	NCT03476798
Niraparib	Copanlisib	Phase I	Active	NCT03586661
Niraparib	Dostarlimab	Phase II	Not yet recruiting	NCT05870761
Olaparib or AZD6738	AZD6738 or Durvalumab	Phase II	Recruiting	NCT03682289
Olaparib	DS-8201a	Phase I	Recruiting	NCT04585958
BBI-355	Monotherapy	Phase I	Recruiting	NCT05827614
BAY1895344	Chemotherapy	Phase I	Recruiting	NCT04491942
ART0380	Monotherapy	Phase II	Not yet recruiting	NCT05798611
AZD1775	Radiotherapy and chemotherapy	Phase I	Active	NCT03345784

## Data Availability

All data generated or analyzed during this study are included in this published article.

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
