# Peer review of "Targeting DNA Damage Repair and Immune Checkpoint Proteins for Optimizing the Treatment of Endometrial Cancer"

_pharmaceutics, 2023, doi:10.3390/pharmaceutics15092241_

Round 1

Reviewer 1 Report

The review of DNA damage repair and immune checkpoints in the context of endometrial cancer (EC) management done by Bian et al is a very-well composed article that arouses general interest and adds largely to the existing knowledge.

The authors have very well researched the available articles going back to the 1980s and compiled the literature on the current perspectives for such a subject that has been studied for a very long time. The Introduction has nicely summarized the background and molecular profiling of EC and continued to provide the rationale of the available treatments targeting DNA damage repair landmarks, e.g., PARP. The motivation of the review on the clinical and preclinical studies on DNA damage-related targets as well as immunotherapy is also apt. There is a general gap in the available knowledge and the article attracts interest in this regard.

Sections 2, 3, 4 – DNA Damage Repair Mechanism – This section describes the various mechanisms to counteract DSB or SSB. However, it is to be noted that the main goal of this article is to educate the scientific community on the available treatment strategies to fight DNA damage. Since the authors have already shown the major DNA damage repair strategies in the figures 1, 2 & 3 (which provides great relevance to the text), the text in these sections is more than necessary and it takes away the main interest from the rest of the manuscript. In my humble opinion, these sections need to be reduced (maybe to half of the original) making them only necessary to shortly describe the mechanisms followed by available treatment strategies discussed in the subsequent sections.

Sections 5, 6, and 7 are the stars of this study and the authors have done a fantastic job here. The Conclusion should be reduced and only focused on the pros of targeting DNA damage response pathways. Parts of it, especially the limitations can be shifted to the next section with the future perspectives.

There are some grammatical and syntax errors in the manuscript. Overall copyediting of the manuscript to increase its readability from a professional service is highly recommended.

Author Response

Response to Reviewer 1 Comments

Point 1: Sections 2, 3, 4 – DNA Damage Repair Mechanism – This section describes the various mechanisms to counteract DSB or SSB. However, it is to be noted that the main goal of this article is to educate the scientific community on the available treatment strategies to fight DNA damage. Since the authors have already shown the major DNA damage repair strategies in the figures 1, 2 & 3 (which provides great relevance to the text), the text in these sections is more than necessary and it takes away the main interest from the rest of the manuscript. In my humble opinion, these sections need to be reduced (maybe to half of the original) making them only necessary to shortly describe the mechanisms followed by available treatment strategies discussed in the subsequent sections.

Response 1: Thank you very much for the helpful comments. We re-wrote these sections to accurately describe mechanisms followed by available treatment strategies, please see the revised manuscript.

Point 2: Sections 5, 6, and 7 are the stars of this study and the authors have done a fantastic job here. The Conclusion should be reduced and only focused on the pros of targeting DNA damage response pathways. Parts of it, especially the limitations can be shifted to the next section with the future perspectives.

Response 2: Thank you for the constructive suggestion. We have reduced the conclusion that only focused on the pros of targeting DNA damage response pathways. Furthermore, we have moved the limitations in future perspectives section.

Point 3: There are some grammatical and syntax errors in the manuscript. 

Response 3: Thank you very much for the helpful comments. We apologize for the errors about the grammatical and syntax. We have rechecked the manuscript and corrected the grammar errors as much as we can. We believe that the revised manuscript is much improved thanks to your helpful comments.

Reviewer 2 Report

In this review article, the authors summarized the therapeutic potential of targeting DDR proteins as monotherapy or in conjunction with cytotoxic drugs for the treatment of EC, and the involvement of the DNA repair mechanism in the initiation, progression, and maintenance of the disease. Finally, the authors summarized new immunotherapies for advanced EC that have been tested or are undergoing testing, and suggest future avenues for the development of DDR-based cancer treatments for the treatment of EC.

The topic of this review is interesting and could have a significant contribution to the field of EC therapy. I only have some minor concerns.

1) Line 11: Delete “of”

2) L31: Change to “diabetes [4,5].”

3) L38 and L58: Explain “endometrioid” What is the difference between endometrioid EC and EC

4) In all figures cite the software that was used to illustrate figures. Also add suitable references.

5) Cite Figures 2 - 4 in the appropriate places in the text.

6) L245: “olaparib” however in L246 “Olaparib” please unify the name of all drugs throughout the whole manuscript.

7) “5. Targeting DNA Damage Repair Proteins to Treat Endometrial Cancer” Add a table to summarize the text underneath this subtitle.

8) More details are required in “6. The DNA Damage Response Correlates with Cancer Cell Immunogenicity”

9) L383: “findings” is not appropriate here, replace with another suitable word.

10) L393: “of studies” delete “of”

11) Many abbreviations were found in this review so I suggest a list of abbreviations.

12) L428: “Food and 428 Drug Administration. Accessed December 20, 2020.https://www.accessdata.fda.gov/drugsatfda_docs/label/2017/125514s014lbl.pdf )” What is this? Is it a reference? Please check and correct.

13) It is better to tubulate different subtitles of “7. Exploiting the Interplay of DNA Damage Response and Immunity for Cancer Therapy” to make it easy for readers.

Only some minor errors, see my comments.

Author Response

Response to Reviewer 2 Comments

The topic of this review is interesting and could have a significant contribution to the field of EC therapy. I only have some minor concerns.

Point 1: Line 11: Delete “of”

Response 1: Thank you very much for the suggest. We have deleted “of” in the revised manuscript.

Point 2: Change to “diabetes [4,5].”

Response 2: Thank you very much for the suggest. We have changed to “diabetes [4,5].” in the revised manuscript.

Point 3: L38 and L58: Explain “endometrioid” What is the difference between endometrioid EC and EC

Response 3: In response to the reviewer’s comments, we delineated the question as follow: EC includes type I and type II EC, most of type I ECs are belong to endometrioid EC while most of type II ECs are display with serous histology.

Point 4: In all figures cite the software that was used to illustrate figures. Also add suitable references.

Response 4:  Thank you for the helpful comments.  We used PPT software to delineate the figures.

Point 5: Cite Figures 2 - 4 in the appropriate places in the text.

Response 5: Thank you for the helpful suggestion. We have cited Figures 2 - 4 in the appropriate places in the revised manuscript.

Point 6: “olaparib” however in L246 “Olaparib” please unify the name of all drugs throughout the whole manuscript.

Response 6: We apologize for the inconsonant writing and have now unified the name of all drugs in the revised manuscript.

Point 7: “5. Targeting DNA Damage Repair Proteins to Treat Endometrial Cancer” Add a table to summarize the text underneath this subtitle.

Response 7: Thank you for the helpful comments. We have added a table regarding the DDR inhibitor that tested in the clinic for the treatment of EC in the revised manuscript.

Point 8: More details are required in “6. The DNA Damage Response Correlates with Cancer Cell Immunogenicity”

Response 8: We thank the Reviewer for the thoughtful comments. We have rewritten this section in the revised manuscript.

Point 9: L383: “findings” is not appropriate here, replace with another suitable word.

Response 9: Thank you for the constructive suggestion. We have rewritten this sentence in the revised manuscript.

Point 10:  L393: “of studies” delete “of”

Response 10: Thank you for the helpful comments. We have deleted “of” in the revised manuscript.

Point 11: Many abbreviations were found in this review so I suggest a list of abbreviations.

Response 11: Thank you for the helpful suggestion. We have added an abbreviation list in the last section of the manuscript, please find it in the revised manuscript.

Point 12: L428: “Food and 428 Drug Administration. Accessed December 20, 2020.https://www.accessdata.fda.gov/drugsatfda_docs/label/2017/125514s014lbl.pdf )” What is this? Is it a reference? Please check and correct.

Response 12: We apologize for the misleading writing and have now cited a reference in the revised manuscript.

Point 13: It is better to tubulate different subtitles of “7. Exploiting the Interplay of DNA Damage Response and Immunity for Cancer Therapy” to make it easy for readers.

Response 13: Thank you for the helpful suggestion. We have added a treatment list about immune check point inhibitors as monotherapy for the treatment of EC in the revised manuscript.

Reviewer 3 Report

Here Bian and colleagues summarize in detail DNA repair mechanisms in general as well as in the context of endometrial cancer (EC) followed by mainly preclinical studies in EC with related agents.

After a far briefer summary of immunotherapy (IO) and its possible interaction with DNA repair, a largely clinical summary of the various IO agents in EC is given inclduing response relative to DNA repair biomarkers.

Different areas of the paper have merit but the overall integration and effect is somewhat patchy and incongruous with high detail in some areas (4 pages on basic biology of DNA repair) and little in others.

Specific comments by section are:

Significant English editing is required although the overall 'science sense' is reasonable.

Abstract

A review of DNA repair systems in endometrial cancer causation and progression.

I note that the abstract doesn’t talk of interaction between IO and DDR pRx, it focuses almost entirely on DDR with immunotherapy as a seeming afterthought in the final sentence. Suggest a more balanced abstract and possibly mention combination. Otherwise point of having the two areas in this paper are unclear.

Additionally the introduction finishes by talking about identifying more potential targets in EC. You need to decide what this review is about, and cover all proposed abstracts evenly in abstract and introduction.

1.       Introduction

31 ‘menarche’ – do you mean early menarche?

36 ‘serious’ – think you mean ‘serous’. This is mentioned several times

41 ‘inevitably occur’ – is it absolutely inevitable or just very common.

51 ‘classified endometrial cancer’ – are these 4 sub-types all type II ECs? It is not clear.

60 – 73 – the introduction to DDR targeting is somewhat confusing, both due to needing English editing as well as general flow.

74 again mention of IO thrown in at the last minute.

2 – 4 – DNA repair mechanisms

Although the English requires editing, this is a detailed section. However, it is 4 pages of mechanistic science on DNA repair but with little or no reference to cancer along the way. This seems excessive and could almost be a mini-review by itself.

5. Targeting PARP

This section tends to fluctuate between cell line studies to patient case reports and back. The subsequent PARP inhibitor combinations section is better organised.

Suggsest a table of clinical studies, both done and planned even if they are small and few (which is the sense I get here).

311 – 320 ATR-CHK1 signaling – the explanation here is a little complex and hard to understand, introduces some unexplained vocabulary.

Are there any clinical studies targeting ATR-CHK1, there is no mention in this section.

5.3 Targeting WEE1 – early clinical studies are mentioned. What is the next step, are later phase studies planned or in progress?

Immunotherapy introduction - There is a notable imbalance between several pages of DDR science and almost none for immunotherapy which feels out of balance.

The summaries of the various immunotherapies in bladder cancer are pretty good with evidence of efficacy in the context of DNA repair also extracted and given. A table of each therapy summarizing the trials , numbers of EC patients, responses and DNA repair data would give a more clear again view of the activity here.

The combination of immunotherapy with either DNA repair inhibiting agents or chem (although having a rationale) reads as a slightly confusing mix. I would separate into two small sections.

Conclusions – 515-525 the first half of this paragraph summarizes EC in general and doesn’t relate to the content of the review. The whole reads more like an introduction than conclusion

Significant English editing is required although the overall 'science sense' is reasonable.

Author Response

Response to Reviewer 3 Comments

Point 1: I note that the abstract doesn’t talk of interaction between IO and DDR pRx, it focuses almost entirely on DDR with immunotherapy as a seeming afterthought in the final sentence. Suggest a more balanced abstract and possibly mention combination. Otherwise point of having the two areas in this paper are unclear.

Additionally the introduction finishes by talking about identifying more potential targets in EC. You need to decide what this review is about, and cover all proposed abstracts evenly in abstract and introduction.

Response 1: We thank the Reviewer for the thoughtful comments. We have rewritten the abstract and introduction to make it more suitable for this manuscript, please see the revised manuscript.

Point 2: 31 ‘menarche’ – do you mean early menarche?

Response 2: Thank you for the helpful comment. We have changed the menarche into early menarche in the revised manuscript.

Point 3: 36 ‘serious’ – think you mean ‘serous’. This is mentioned several times

Response 3: Thank you for the helpful comment. We have now corrected all the ‘serious’ into ‘serous’ in the revised manuscript.

Point 4: 41 ‘inevitably occur’ – is it absolutely inevitable or just very common.

Response 4: Thank you for the thoughtful comment. We have corrected ‘inevitably occur’ into ‘easily occur’ in the revised manuscript.

Point 5: 51 ‘classified endometrial cancer’ – are these 4 sub-types all type II ECs? It is not clear.

Response 5: Thank you for the thoughtful comment. We have now corrected it into “Currently, molecular characteristic studies including whole-exome sequencing and TCGA’s integrated genome sequencing have classified endometrioid and serous histology of EC into four distinct subtypes” in the revised manuscript.

Point 6: 60 – 73 – the introduction to DDR targeting is somewhat confusing, both due to needing English editing as well as general flow.

Response 6: Thank you for the constructive suggestion. We have now rewritten the introduction to make it more suitable for this manuscript, please see the revised manuscript.

Point 7: 74 again mention of IO thrown in at the last minute.

Response 7: We apologize for the shorten writing about the immunotherapy in the introduction section, and we have now rewritten IO in the introduction section to make it more suitable for the manuscript, please see the revised manuscript.

Point 8: 2 – 4 – DNA repair mechanisms

Although the English requires editing, this is a detailed section. However, it is 4 pages of mechanistic science on DNA repair but with little or no reference to cancer along the way. This seems excessive and could almost be a mini-review by itself.

Response 8: Thank you for the constructive suggestion. We have rechecked the manuscript and rewritten the DNA repair mechanisms as much as we can. We believe that the revised manuscript is much improved thanks to your helpful comments.

Point 9: This section tends to fluctuate between cell line studies to patient case reports and back. The subsequent PARP inhibitor combinations section is better organised.

Suggsest a table of clinical studies, both done and planned even if they are small and few (which is the sense I get here).

Response 9: We thank for the reviewer’s constructive suggestion. We have now added a table of clinical studies in the revised manuscript.

Point 10: 311 – 320 ATR-CHK1 signaling – the explanation here is a little complex and hard to understand, introduces some unexplained vocabulary.

Are there any clinical studies targeting ATR-CHK1, there is no mention in this section.

Response 10: Thank you for the thoughtful comment and constructive suggestion. We have now rewritten the ATR-CHK1 signaling to make it easily readable in the revised manuscript. We also added a table of clinical studies in the revised manuscript.

Point 11: 5.3 Targeting WEE1 – early clinical studies are mentioned. What is the next step, are later phase studies planned or in progress?

Response 11: Thank you for the helpful comment. We have added a table of clinical studies of WEE1 inhibitor in the treatment of EC in the revised manuscript.

Point 12: The summaries of the various immunotherapies in bladder cancer are pretty good with evidence of efficacy in the context of DNA repair also extracted and given. A table of each therapy summarizing the trials, numbers of EC patients, responses and DNA repair data would give a more clear again view of the activity here.

Response 12: We thank for the reviewer’s constructive suggestion. We have now added a table of Clinical studies of ICIs as monotherapy in EC in the revised manuscript.

Point 13: The combination of immunotherapy with either DNA repair inhibiting agents or chem (although having a rationale) reads as a slightly confusing mix. I would separate into two small sections.

Response 13: We thank for the reviewer’s constructive suggestion. We have separated this section into two small sections, please find it in the revised manuscript.

Point 14: Conclusions – 515-525 the first half of this paragraph summarizes EC in general and doesn’t relate to the content of the review. The whole reads more like an introduction than conclusion

Response 14: Thank you for the helpful comment. We have rechecked and rewritten the conclusions as much as we can. We believe that the revised manuscript is much improved thanks to your helpful comments.

Round 2

Reviewer 3 Report

Targeting DNA Damage Repair and Immune Checkpoint Proteins for Optimizing Endometrial Cancer Treatment.

Abstract has better balance.

Last sentence could suggest adding immunotherapy to DDR therapy as that is presumptively the reason you have combined them in the same article.

Intro

Good covering of 2 basic sub-types and drivers.

Line 57 - high-grade or serous-like EC always experience recurrence or tumor progression. I presume you mean they frequently relapse or progress. Seems unlikely that 100% of pts die.

Line 70 – you state that type 1s are usually HR competent, and modest outcomes for DDR inhibition in EC but then state ‘targeting DNA damage repair system in EC will provide a rational way for the treatment of EC independent of different histology types’. This does not appear to follow – surely only Type 2s should be treated this way?

Line 74 – you appear to justify immunotherapy here just because it is useful in lots of cancers, not because (as you say in line 18), ‘recent studies indicate that the application of DNA damage agents in cancer cells leads activation of innate and adaptive immune response, targeting immune checkpoint proteins could overcome immune suppressive environment in tumors.’ Suggest your intro should give a better rationale for reviewing DDR and immunotherapy in the same article (as in line 18).

Line 79 – ‘Lastly, we try to identify more potential targets that can serve as candidates for treating EC.’ This seems like a bit of a random add-on, in an already long review, what is the justification for putting in the same review,?

There is still a long (4 page – line 81 to 205) section that is a very detailed review of DNA repair without any mention of endometrial cancer. I would still suggest reducing this as most people selecting to read are looking for DDR in EC rather than a big general review on DDR.

Considering the size of the review, I would tend to leave case studies out as they are only slightly informative.

Section 5 is useful and more what the paper should involve at an earlier stage, we are on page 7 before a treatment result is mentioned. Table 1 is a good addition but the treatment on each of the arms should be given – it should mention the actual trial agent tested, not just the class and also give the drugs in the control arm specifically. Some controls are mentioned eg carboplatin, others not eg ‘anti-cancer drugs’. Also referencing where any publications in addition to NCT number would be good.

Section 6 and 7 – good section, joins the core concepts of the review together.

Table 2 – again good addition but more detail on treatment arm and any results (noting all have a reference) even though also in text.

Section 7.6 should be section 8 I think as the review is about immunotherapy and DDR therapy – this is the section that brings the two together. Section 7.7 could go before as section 7.6.

Conclusion is only about DDR therapy, no mention of immunotherapy or the combo. Should be in conclusions. Immunotherapy also only briefly mentioned in future directions.

Conclusions and future directions are a bit mixed together. As you have a future directions paragraph, conclusions should just conclude with a summary of results so far, saving comment on future work for the FD paragraph.

Article still needs editing for frequent minor English grammar errors although the general ‘science sense is good’.

EC and endometrial cancer are used in different places in the article – use one or the other.

Author Response

Point 1: Line 57 - high-grade or serous-like EC always experience recurrence or tumor progression. I presume you mean they frequently relapse or progress. Seems unlikely that 100% of pts die.

Response 1: Thank you for the helpful comment. We have rechecked and rewritten this sentence as ‘‘While most of low-grade or intermediate-grade endometrioid EC has a favorable prognosis, the majority of high-grade or serous-like EC always experience recurrence or tumor progression and ultimately succumb to their disease’’. We believe that the revised manuscript is much improved thanks to your helpful comments.

Point 2: Line 70 – you state that type 1s are usually HR competent, and modest outcomes for DDR inhibition in EC but then state ‘targeting DNA damage repair system in EC will provide a rational way for the treatment of EC independent of different histology types’. This does not appear to follow – surely only Type 2s should be treated this way?

Response 2: We thank the Reviewer for the thoughtful comments. While type 1 EC always with intact HR system, most of type 1 ECs are PTEN-deficient. PTEN-deficient ECs are reported sensitive to DDR inhibitors such as PARP inhibitor, besides, combination of DDR inhibitors would hope to extend the efficacy of these inhibitors beyond their respective molecularly defined genetic background. For type 2 EC, a subset of type 2 ECs are HR-deficiency, indicating these patients may benefit from treatments targeting this defect, such as platinum compounds and PARP inhibitors. Together, both type 1 and type 2 ECs would benefit from DDR inhibitors as monotherapy and/or combination therapy.

Point 3: Line 74 – you appear to justify immunotherapy here just because it is useful in lots of cancers, not because (as you say in line 18), ‘recent studies indicate that the application of DNA damage agents in cancer cells leads activation of innate and adaptive immune response, targeting immune checkpoint proteins could overcome immune suppressive environment in tumors.’ Suggest your intro should give a better rationale for reviewing DDR and immunotherapy in the same article (as in line 18).

Response 3: Thank you for the constructive suggestion. We have rechecked the manuscript and rewritten this section as much as we can. We believe that the revised manuscript is much improved thanks to your helpful comments.

Point 4: Line 79 – ‘Lastly, we try to identify more potential targets that can serve as candidates for treating EC.’ This seems like a bit of a random add-on, in an already long review, what is the justification for putting in the same review,?

Response 4: We thank the Reviewer for the thoughtful comments. We have now deleted this sentence in the revised manuscript.

Point 5: There is still a long (4 page – line 81 to 205) section that is a very detailed review of DNA repair without any mention of endometrial cancer. I would still suggest reducing this as most people selecting to read are looking for DDR in EC rather than a big general review on DDR.

Response 5: Thank you for the constructive suggestion. We have now reduced the content of this section as much as we can, we believe that the revised manuscript is much improved thanks to your helpful comments.

Point 6: Considering the size of the review, I would tend to leave case studies out as they are only slightly informative.

Response 6: Thank you for the helpful comment. We have now deleted the citation of case study in the revised manuscript.

Point 7: Section 5 is useful and more what the paper should involve at an earlier stage, we are on page 7 before a treatment result is mentioned. Table 1 is a good addition but the treatment on each of the arms should be given – it should mention the actual trial agent tested, not just the class and also give the drugs in the control arm specifically. Some controls are mentioned eg carboplatin, others not eg ‘anti-cancer drugs’. Also referencing where any publications in addition to NCT number would be good.

Response 7: We thank the Reviewer for the thoughtful comments. We have now added the detailed information of clinical trials of DDR inhibitors as monotherapy or combination therapy tested in EC in the revised table 1, also, as most of the trials are clinical study, so it is hard to cite a reference in the manuscript. We believe that the revised manuscript is much improved thanks to your helpful comments.

Point 8: Section 6 and 7 – good section, joins the core concepts of the review together.

Response 8: Thank you for the helpful comment. The section 6 and section 7 are delineate the association between DNA damage response and anti-tumor immunity, and trying to exploit this interplay for enhancing EC treatment efficiency.

Point 9: Table 2 – again good addition but more detail on treatment arm and any results (noting all have a reference) even though also in text.

Response 9: Thank you for the constructive suggestion. We have now added the detailed information of clinical trials of ICIs as monotherapy tested in EC in the revised table 2.

Point 10: Section 7.6 should be section 8 I think as the review is about immunotherapy and DDR therapy – this is the section that brings the two together. Section 7.7 could go before as section 7.6.

Response 10: Thank you for the constructive suggestion. We have now changed the section 7.6 into section 8 and put the section 7.7 before section 7.6.

Point 11: Conclusion is only about DDR therapy, no mention of immunotherapy or the combo. Should be in conclusions. Immunotherapy also only briefly mentioned in future directions.

Response 11: Thank you for the helpful comment. We have now rewritten the conclusion as much as we can. We believe that the revised manuscript is much improved thanks to your helpful comments.

Point 12: Conclusions and future directions are a bit mixed together. As you have a future directions paragraph, conclusions should just conclude with a summary of results so far, saving comment on future work for the FD paragraph.

Response 12: Thank you for the helpful comment. We have now rewritten the conclusion in the revised manuscript as much as we can. In the FD paragraph, we try to suggest more potential therapeutic strategies such as combination of DDR inhibitors with small molecules targeting epigenetic factors that hope to benefit more EC patients. Furthermore, we give a suggestion of more preclinical studies should focus on utilizing the combination of ICI with DDRi to overcome ICI resistance in the treatment of EC.